# Longstanding smoking associated with frontal brain lobe atrophy: a 32-year follow-up study in women

Lena Johansson ,[1,2,3] Xinxin Guo,[1] Simona Sacuiu ,[1]
Madeleine Mellqvist Fässberg,[1] Silke Kern,[1] Anna Zettergren,[1] Ingmar Skoog[1]

[1]Department of Psychiatry and Neurochemistry, Sahlgrenska Academy, Centre for Ageing and Health (AgeCap), Institute of Neuroscience and Physiology, University of Gothenburg, Goteborg, Sweden
[2]Department of Addiction and Dependency, Sahlgrenska University Hospital, Sahlgrenska universitetssjukhuset, Goteborg, Sweden
[3]Institute of Health and Care Sciences at Sahlgrenska Academy, University of Gothenburg, Gothenburg, Sweden

**Correspondence to**
Dr Lena Johansson;
lena.johansson@neuro.gu.se

## ABSTRACT

**Objective** To examine the association between midlife tobacco smoking and late-life brain atrophy and white matter lesions.

**Methods** The study includes 369 women from the Prospective Population Study of Women in Gothenburg, Sweden. Cigarette smoking was reported at baseline 1968 (mean age=44 years) and at follow-up in 1974–1975 and 1980–1981. CT of the brain was conducted 32 years after baseline examination (mean age=76 years) to evaluate cortical atrophy and white matter lesions. Multiple logistic regressions estimated associations between midlife smoking and late-life brain lesions. The final analyses were adjusted for alcohol consumption and several other covariates.

**Results** Smoking in 1968–1969 (adjusted OR 1.85; 95% CI 1.12 to 3.04), in 1974–1975 (OR 2.37; 95% CI 1.39 to 4.04) and in 1980–1981 (OR 2.47; 95% CI 1.41 to 4.33) were associated with late-life frontal lobe atrophy (2000–2001). The strongest association was observed in women who reported smoking at all three midlife examinations (OR 2.63; 95% CI 1.44 to 4.78) and in those with more frequent alcohol consumption (OR 6.02; 95% CI 1.74 to 20.84). Smoking in 1980–1981 was also associated with late-life parietal lobe atrophy (OR 1.99; 95% CI 1.10 to 3.58). There were no associations between smoking and atrophy in the temporal or occipital lobe, or with white matter lesions.

**Conclusion** Longstanding tobacco smoking was mainly associated with atrophy in the frontal lobe cortex. A long-term stimulation of nicotine receptors in the frontal neural pathway might be harmful for targeted brain cell.

## INTRODUCTION

Cigarette smoking continues to be one of the most common addictive habits globally and is the single greatest cause of preventable death.[1] Tobacco consists of thousands of harmful compounds, of which many are directly toxic to the brain and the central nervous system (CNS).[2] The short-term effect on the brain is rather well known, but less research has been done regarding the long-term consequences.

Brain lesions increase with age, often without an identified aetiology.[3 4] Cortical

---

atrophy is due to advanced loss of neurons in grey matter areas, expressed either generally in the entire cortex, or partial, in specific brain areas. The neurodegeneration can be a result of proteopathies, inflammatory processes or oxidative stress.[4] Subcortical lesions are most often related to cerebrovascular damages, such as small-vessel disease or brain infarcts, and relate to a widespread damage on the cell axons in white matter areas.[5] The consequences of extended cortical and subcortical shrinkage can be expressed as cognitive and functional impairment.[4]

### STRENGTHS AND LIMITATIONS OF THIS STUDY

⇒ Analyses were done in a prospective population-based dataset with repeated information on smoking habits and other variables during participants' middle age.
⇒ There were no available data on the number of cigarettes/per day, smoking habits before and the study period, or other forms of nicotine use.
⇒ It can be a potential stigma related to the self-reporting of cigarettes and alcohol use.
⇒ Visual rating of brain atrophy and white matter lesions on CT is a rather crude method, compared with, for example, MRI.
⇒ CT data from earlier in life were not available. Cortical atrophy in midlife could have led to addiction habits such as smoking, that is, reverse causality.

A number of genetic and life-style factors have been associated with structural brain changes.[6–10] Tobacco use poses a risk to the brain due to the exposure of toxic chemicals, but few studies have examined long-term associations between cigarette smoking and late-life brain lesions.[11–13] The most addictive and intoxicating ingredient in tobacco is nicotine, an alkaloid that mainly binds to acetylcholine receptors in CNS,[14] and interacts with the dopaminergic reward system.[2 15] Chronic tobacco smoking increases the amount of nicotine receptors in the brain and changes the structure and function of involved

neurons.[2] Nicotine has also negative vascular effects to the brain, due to arteriolosclerosis and increased blood pressure.[2 16]

In this study, we analysed the relation between midlife tobacco smoking and the presence of grey matter atrophy and white matter lesions on CT, in late-life, in a population-based sample of 369 women followed over 32 years. We had an extra focus on alcohol consumption, given its high comorbidity with smoking and emerging findings regarding their interactions.[17] The study adjusts for several important covariates, for example, vascular risk factors.

## METHODS
### Study population
The study is part of the Prospective Population Study of Women in Gothenburg which was initiated in 1968 with a comprehensive examination of 1462 women (participation rate 90%) born in 1908, 1914, 1918, 1922 and 1930.[18 19] Individuals were systematically sampled from the Swedish Population Register, based on specific birth dates, to yield a representative sample. Follow-ups were performed in 1974–1975, 1980–1981, 1992–1993 and 2000–2001, with participation rates among survivors of 91%, 83%, 70% and 71%, respectively. In 2000–2001, all participants (n=684) were invited for CT scan of the brain examination. Participants and non-participants in the CT study (n=379 vs n=305, participation rate 55%) were similar regarding tobacco smoking, diastolic and systolic blood pressure, serum cholesterol, body mass index (BMI) and frequency of stroke and myocardial infarction. However, CT participants were younger and had lower 5-year mortality rate compared with those who declined to take part.

The current study includes 369 women, who responded to the question about cigarette smoking in any of the three midlife examinations, 1968–1969, 1974–1975 and 1980–1981 and took part in the CT study in 2000–2001 (10 women were excluded due to missing information on smoking habits). Informed consent was obtained from participants and/or their relatives.

### Assessment of smoking habits
A question about tobacco smoking habits was asked and the alternative answers were: 0=never smoked; 1=smoked earlier, but not during the last 15 years; 2=stopped smoking 1–15 years ago; 3=stopped smoking 0–12 months ago; 4=current smoking, but no deep drag; or 5=current smoker, does deep drag. In this study, we used 3–5 to define a 'current smoker'.

### Assessment of CT scans of the brain
CT scans of the brain were performed without contrast enhancement and with 10 mm contiguous slices. A neurologist experienced in visual CT rating evaluated all images.[20] Cortical atrophy of the temporal, frontal, parietal and occipital lobes was categorised according to the anatomical subdivision. Severity was scored as normal, mild and moderate–severe, according to the extent of sulcal widening.[20 21] White matter lesions were defined as low-density areas in the periventricular subcortical white matter, and decreased density was rated as mild, moderate and severe in relation to the attenuation of normal white matter.[20–24] The examiner was blinded to the participants' clinical characteristics.

An interobserver agreement between the rating neurologist and a neuroradiologist was done in 130 scans.[20] The inter-rater kappa value (κ) was 0.61 for temporal lobe atrophy (78% concordance), 0.53 for frontal lobe atrophy (74% concordance), 0.50 for parietal lobe atrophy (80% concordance), 0.62 for occipital lobe atrophy (90% concordance) and 0.65 for presence and severity of white matter lesions (82% concordance), that is, an inter-rater reliability between moderate (κ=0.50) and substantial (κ=0.65).[20]

### Assessment of other measurements
Information on alcohol consumption, body length, body height, serum cholesterol, blood pressure, antihypertensive medication, physical activity and psychological stress were obtained at each midlife examination. The use of alcohol consumption (beer, wine and/or spirits) was categorised into two groups, that is, less than or equal to once weekly and more than once weekly. Body height was measured to the nearest centimetre and weight to the nearest 0.1 kg. BMI was calculated using the formula kg/m$^2$. Blood samples were taken after an overnight fast, and serum cholesterol concentrations were measured. Blood pressure was measured in the sitting position after 5 min' rest. Hypertension was defined as systolic blood pressure≥150 mm Hg and/or diastolic blood pressure≥90 mm Hg and/or taking antihypertensive medication. Psychological stress was defined as several periods of stress symptoms (eg, tension, nervousness or sleep disturbance) during the last 5 years. Physical activity during leisure time was rated as low (<4 hours/week) and medium/high (≥4 hours/week).

Information on stroke, myocardial infarction, heart failure and diabetes was obtained from the Swedish Hospital Discharge Register, and any of these diagnoses before 31 December 2002 were defined as cardiovascular and cerebrovascular disease (CCVD). Dementia was diagnosed according to the Diagnostic and Statistical Manual of mental disorders (DSM) III-R criteria, based on information from the psychiatric examinations, close informant interviews, medical records and the Swedish Hospital Discharge Registry, during the entire study period, that is, from baseline in 1968 to 31 December 2002.[24–26]

### Statistical methods
Independent sample t-tests or $\chi^2$ tests were used to compare CT participants and non-CT participants. Logistic regression analyses estimated associations between report of midlife smoking, in 1968–1981, and cortical atrophy

(no vs mild–severe) and periventricular white matter lesions (no vs mild–severe) at CT examination in 2000–2001. The associations are presented as OR and 95% CI, adjusted for age in first model. In a second model, we adjusted for age, alcohol consumption, BMI, cholesterol, hypertension, psychological stress, low physical activity and CCVD. Persons with missing data in the BMI (n=1) and cholesterol (n=1) variables were given the mean level of the variable, and missing data in the alcohol consumption, hypertension (n=17) and psychological stress (n=8) variables were rated as '0' (not fulfilled criteria). Hence, all participants were included in both models.

We further examined whether the number of examinations when smoking was reported influenced the associations to brain atrophy and white matter lesions. The study sample was reclassified into three groups: (1) *never smokers*=never reported smoking at any examination, (2) *temporary smokers*=report of smoking at one or two examinations and (3) *longstanding smokers*=report of smoking at all three midlife examinations. The association between longstanding smoking and frontal lobe atrophy was analysed in two logistic regression models; in the first model, we adjusted for all described covariates, and in the second model we adjusted for white matter lesions and dementia. Analyses were studied separately in the later-born cohorts (born in 1922 and 1930) and the earlier-born cohorts (born in 1908, 1914 and 1918).

Finally, age-adjusted logistic regressions analysed the association between alcohol consumption, longstanding smoking and frontal lobe atrophy. An interaction analysis was done to evaluate the interaction between smoking and alcohol consumption (longstanding smoking×more frequent alcohol consumption) in relation to frontal lobe atrophy (significance level p<0.20). The sample was stratified by alcohol consumption (lower vs higher), and we reanalysed the association between smoking and frontal lobe atrophy using logistic regressions within each stratum. Correlations between the five outcome variables (cortical atrophy and white matter lesions) were tested and reported by Pearson correlation coefficient (95% CI), and the p values, of main findings, were multiplied with five (numbers of outcomes) to control for false positive result in multiple analyses. All analyses were performed with IBM SPSS V.24 statistical software.

### Patient and public involvement

Patients and/or the public were not involved in the design, or conduct, or reporting, or dissemination plans of this research.

### RESULTS

Characteristics of the study sample (n=369) are shown in table 1. The mean age at the baseline examination 1968–1969 was 44 years (SD±6 years) and cigarette smoking was reported by 32% (n=119) in 1968–1969, 29% in 1974–1975 and 25% in 1980–1981. Overall, 16% in earlier-born cohorts (1908, 1914 and 1918), and 38% in the later-born

**Table 1** Characteristics of the study sample (n=369)

| | |
|---|---|
| **Baseline age (years), mean±SD** | **43.8±5.6** |
| Birth year 1908, n (%) | 2 (0.5) |
| Birth year 1914, n (%) | 19 (5.1) |
| Birth year 1918, n (%) | 80 (21.7) |
| Birth year 1922, n (%) | 105 (28.5) |
| Birth year 1930, n (%) | 163 (44.2) |
| Current smoking, n (%) | |
| Examination 1968–1968 | 119 (32.2) |
| Examination 1974–1975 | 104 (28.8) |
| Examination 1980–1981 | 88 (25.0) |
| Covariates | |
| Alcohol consumption (more than once weekly)*, n (%) | 101 (27.4) |
| Body mass index*†, mean±SD | 23.46±0.17 |
| Cardiovascular and cerebrovascular diseases§, n (%) | 53 (14.4) |
| Cholesterol (mmol/L)*†, mean±SD | 6.59±0.08 |
| Hypertension*§, n (%) | 77 (20.9) |
| Psychological stress*‡, n (%) | 67 (18.2) |
| Low physical activity (<4 hours/week)*, n (%) | 51 (13.8) |
| Dementia,† n (%) | 17 (4.6) |

*Measured at baseline, 1968.
†One with missing information.
‡Seventeen with missing information.
§Measured until 31 December 2002.
¶Eight with missing information.

cohorts (1922 and 1930), reported smoking in baseline (table 2). Participants data from the CT study in 2000–2001 (mean age 76 years SD±6 years) are presented in table 3. All outcome variables, atrophy and white matter lesions were associated to each other, according to Pearson correlation coefficient (see online supplemental file 1).

Cigarette smoking in 1968–1969 (adjusted OR 1.85; 95% CI 1.12 to 3.04), 1974–1975 (OR 2.37; 95% CI 1.39 to 4.04) and 1980–1981 (OR 2.47; 95% CI 1.41 to 4.33) was associated with late-life frontal lobe atrophy (table 4). Report of smoking in the last examination (1980–1981) was also associated with parietal lobe atrophy (OR 1.99; 95% CI 1.10 to 3.58). There was no association between smoking and cortical atrophy in the temporal or occipital lobes, or with periventricular white matter lesions (table 4).

Among the 351 women who participated in all 3 midlife examinations, 65% were never-smokers, 12% reported smoking at 1 or 2 examinations and 23% reported smoking at all 3 examinations (table 5). The longstanding smokers (report of smoking in all three examinations) had a higher prevalence of late-life frontal lobe atrophy compared with never-smokers (adjusted OR 2.63; 95% CI 1.44 to 4.78, p 0.003). The p value was further multiplied

**Table 2** Prevalence of cigarette smoking in midlife examinations, presented by birth cohort

|  | Born in 1908 | Born in 1914 | Born in 1918 | Born in 1922 | Born in 1930 |
|---|---|---|---|---|---|
| Current smoking, 1968–1969, n (%) | 0 (0) | 2 (10.5) | 16 (20.0) | 38 (36.2) | 63 (38.7) |
| Currant smoking, 1974–1975, n (%) | 0 (0) | 2 (10.5) | 14 (18.2) | 28 (26.9) | 60 (37.7) |
| Current smoking, 1980–1981, n (%) | 0 (0) | 2 (10.5) | 11 (14.9) | 23 (22.8) | 52 (33.3) |

with five (numbers of outcomes) to control for false positive result in multiple analyses and were still significant (p 0.003×5=p 0.015), and the association also remained after additional adjusting for white matter lesions (OR 2.57, 95% CI 1.41 to 4.69). There were no associations between temporary smoking (smoking in 1–2 examination) and frontal lobe atrophy, nor between longstanding smoking and lesions in the other cortical and white matter areas (table 5). The adjusted ORs (95% CI) between longstanding smoking and frontal lobe atrophy were 5.31 (0.59 to 47.84) in the earlier-born cohorts (born in 1908, 1915 and 1918) and 2.49 (1.31 to 4.74) in the later-born cohorts (born in 1922 and 1930).

There was an interaction between smoking and alcohol consumption (longstanding smoking×higher alcohol consumption) in relation to frontal lobe atrophy (age-adjusted $p_{interaction}$ 0.13). The sample was thus stratified to explore this interaction in more detail. In women with a more frequent alcohol consumption (more than once weekly, n=101), longstanding smoking was associated with frontal lobe atrophy (adjusted OR 6.02, 95% CI 1.74 to 20.84), while no such association was found in those with low alcohol consumption (p>0.05). In addition, a more frequent alcohol consumption was associated with longstanding smoking (p 0.02) but not with frontal lobe atrophy (p 0.63).

In total, 17 women developed dementia (Alzheimer's disease or vascular dementia) during the study period between 1968 and 2001. The association between longstanding smoking and frontal lobe atrophy remained after additional adjustment for dementia (OR 2.50; 95% CI 1.38 to 4.56).

## DISCUSSION

In this population-based sample of women followed over 32 years, we found that midlife tobacco smoking was associated with late-life frontal lobe atrophy. The strongest association was observed in those who reported smoking in all three midlife examinations and in women with a more frequent alcohol consumption. There was also an association between smoking in the last midlife examination (1980–1981) and parietal lobe atrophy. Being a temporary smoker was not associated with late-life brain lesions. There was neither association between smoking and atrophy in other cortical areas nor in the white matter. The findings remained after controlling for several potential confounders including lifestyle factors, vascular diseases and dementia.

Our findings are difficult to compare with other studies, as few earlier studies have examined the effects of smoking on structural brain lesions on long follow-up periods of decades. Previous studies have mostly been conducted in cross-sectional samples of young and middle-aged people or were based on retrospective information on smoking habits. Among studies in younger populations, a meta-analysis of seven voxel-based morphometry (VBM) studies (n=418, median age 37) reported that chronic smokers had decreased grey matter volume in the anterior cingulate cortex,[12] that is, the area surrounding the frontal part of corpus callosum.[27] Another meta-analysis, based on 11 VBM studies (n=1710, mean age 42 years), established that smokers had smaller grey matter volumes in the prefrontal cortex and larger volumes in lingual cortex, compared with non-smokers.[13] In two recent published

**Table 3** Prevalence of cortical atrophy and white matter lesions on CT scans of the brain in 2000–2001, presented by birth cohort

|  | Born in 1908* (n=2) | Born in 1914† (n=19) | Born in 1918‡ (n=80) | Born in 1922§ (n=105) | Born in 1930¶ (n=163) | Total sample (n=369) |
|---|---|---|---|---|---|---|
| Temporal lobe atrophy, n (%) | 2 (100) | 17 (89.5) | 57 (71.3) | 57 (54.3) | 52 (31.9) | 185 (50.1) |
| Frontal lobe atrophy, n (%) | 1 (50) | 17 (89.5) | 57 (71.3) | 62 (59.0) | 60 (36.8) | 197 (53.4) |
| Parietal lobe atrophy, n (%) | 1 (50) | 15 (78.9) | 42 (52.5) | 49 (46.7) | 27 (16.6) | 134 (36.3) |
| Occipital lobe atrophy, n (%) | 1 (50) | 12 (63.2) | 33 (41.3) | 27 (25.7) | 10 (6.1) | 83 (22.5) |
| White matter lesions, n (%) | 2 (100) | 15 (78.9) | 56 (70.0) | 73 (69.5) | 66 (40.5) | 212 (57.5) |

*Aged 92–93 years.
†Aged 86–87 years.
‡Aged 82–83 years.
§Aged 78–79 years
¶Aged 70–71 years

**Table 4** Associations between midlife smoking and late-life structural brain changes on CT scans in 2000

| | Tobacco smoking in 1968 (n=119) | Tobacco smoking in 1974 (n=104) | Tobacco smoking in 1980 (n=88) |
|---|---|---|---|
| **Temporal lobe atrophy** | | | |
| Cases, n | 53 | 50 | 45 |
| $OR_1$ (95% CI) | 0.93 (0.58 to 1.49) | 1.23 (0.75 to 2.03) | 1.53 (0.90 to 2.61) |
| $OR_2$ (95% CI) | 0.95 (0.58 to 1.56) | 1.27 (0.75 to 2.13) | 1.60 (0.92 to 2.79) |
| **Frontal lobe atrophy** | | | |
| Cases, n | 69 | 63 | 54 |
| $OR_1$ (95% CI) | **1.80 (1.11 to 2.92)** | **2.24 (1.34 to 3.73)** | **2.30 (1.34 to 3.95)** |
| $OR_2$ (95% CI) | **1.85 (1.12 to 3.04)** | **2.37 (1.39 to 4.04)** | **2.47 (1.41 to 4.33)** |
| **Parietal lobe atrophy** | | | |
| Cases, n | 42 | 36 | 34 |
| $OR_1$ (95% CI) | 1.32 (0.80 to 2.19) | 1.37 (0.81 to 2.34) | **1.89 (1.07 to 3.33)** |
| $OR_2$ (95% CI) | 1.38 (0.81 to 2.34) | 1.37 (0.79 to 2.39) | **1.99 (1.10 to 3.58)** |
| **Occipital lobe atrophy** | | | |
| Cases, n | 24 | 21 | 20 |
| $OR_1$ (95% CI) | 1.21 (0.67 to 2.20) | 1.35 (0.73 to 2.55) | 1.75 (0.90 to 3.39) |
| $OR_2$ (95% CI) | 1.25 (0.67 to 2.32) | 1.35 (0.70 to 2.61) | 1.84 (0.92 to 3.67) |
| **White matter lesions** | | | |
| Cases, n | 68 | 61 | 52 |
| $OR_1$ (95% CI) | 1.26 (0.79 to 2.02) | 1.41 (0.86 to 2.32) | 1.51 (0.89 to 2.56) |
| $OR_2$ (95% CI) | 1.43 (0.74 to 2.75) | 1.83 (0.94 to 3.57) | 1.27 (0.62 to 2.64) |

Logistic regression analyses; OR (95% CI).
$OR_1$ adjusted for age.
$OR_2$ adjusted for age, alcohol consumption, body mass index, cardiovascular and cerebrovascular disease, cholesterol, hypertension, physical activity and psychological stress.
The bold values are statistical significant, p-value <0.05

studies, it was found that current smokers had smaller frontal and temporal grey matter volume (n=698, mean age 50 years),[28] and thinning in the orbitofrontal cortex, insula and entorhinal-middle temporal regions (n=82, mean age 45 years).[29] Taken together, less grey matter substance, in various brain regions, has been observed in chronic smokers, and most consistently in the prefrontal and frontal brain areas.

Several biological mechanisms may explain our findings regarding frontal lobe atrophy. Tobacco smoking affects the addiction-related regions in the brain,[28 30] and the frontal brain contains the highest densities of nicotinic acetylcholine receptors.[14] Nicotine interacts with the dopaminergic reward system in the forebrain,[2 15] that is, the mesolimbic system,[14 31 32] which connects the midbrain, with the basal ganglia and the frontal cortex.[13] An acute nicotine intake causes activation of nicotinic receptors, while a chronic use can lead to desensitisation of these receptors, due to the development of tolerance.[13] Therefore, a long-term activation and exhaustion of synapses might lead to receptor damage and neurodegeneration.[14] Data from animal models show that the neurotoxic effects of nicotine can lead to cell loss and synaptic alterations.[33 34] The fact that the mesolimbic

pathway is part of the larger mesocortical pathway might also explain the associations to parietal lobe atrophy,[15] which findings from this study support.

Interestingly, the association between smoking and frontal lobe atrophy varied depending on alcohol habits, that is, women with a more frequent alcohol consumption had more frontal brain atrophy. Similar findings have been observed in earlier studies,[35] which suggests interaction effects between exposures linked to addiction behaviours.[2 15] It is a well-known fact that smoking co-occurs with higher alcohol consumption,[36] and findings from studies on alcohol disorders have shown an intricate interplay between smoking use and alcohol use.[29] Alcohol has also independently been associated with several negative brain outcomes, including smaller frontal volume and desensitisation in the mesolimbic system.[8 37]

We did not find an association between smoking and white matter lesions. Cigarette smoking gives exposure to carbon monoxide, free radicals and free oxygen species, which have negative cerebrovascular effects and increase the risks of arteriolosclerosis, ischaemic demyelination and lower blood flow.[2 16 28] The frontal cortex is vulnerable to vascular diseases, due to its reliance on multiple penetrating arteries in the underlying white matter, with

Table 5 Associations between midlife smoking in multiple examinations, between 1968 and 1980, and late-life structural brain changes on CT scans in 2000 (n=351)

| | No tobacco smoking in any examination (n=227) | Tobacco smoking in 1–2 examination (n=43) | Tobacco smoking in 3 examinations (n=81) |
|---|---|---|---|
| Temporal lobe atrophy | | | |
| Cases | 121 | 17 | 40 |
| $OR_1$ (95% CI) | 1.0 (ref.) | 0.74 (0.37 to 1.51) | 1.24 (0.71 to 2.17) |
| $OR_2$ (95% CI) | 1.0 (ref.) | 0.76 (0.37 to 1.56) | 1.28 (0.72 to 2.30) |
| Frontal lobe atrophy | | | |
| Cases, n | 118 | 19 | 51 |
| $OR_1$ (95% CI) | 1.0 (ref.) | 0.96 (0.48 to 1.91) | **2.36 (1.36 to 4.18)** |
| $OR_2$ (95% CI) | 1.0 (ref.) | 1.00 (0.49 to 2.04) | **2.63 (1.44 to 4.78)** |
| Parietal lobe atrophy | | | |
| Cases, n | 84 | 12 | 31 |
| $OR_1$ (95% CI) | 1.0 (ref.) | 0.94 (0.44 to 2.01) | 1.69 (0.94 to 3.03) |
| $OR_2$ (95% CI) | 1.0 (ref.) | 0.99 (0.45 to 2.18) | 1.82 (0.98 to 3.39) |
| Occipital lobe atrophy | | | |
| Cases, n | 55 | 7 | 18 |
| $OR_1$ (95% CI) | 1.0 (ref.) | 0.99 (0.39 to 2.50) | 1.57 (0.79 to 3.12) |
| $OR_2$ (95% CI) | 1.0 (ref.) | 0.96 (0.37 to 2.46) | 1.60 (0.77 to 3.32) |
| White matter lesions | | | |
| Cases, n | 131 | 22 | 49 |
| $OR_1$ (95% CI) | 1.0 (ref.) | 0.97 (0.49 to 1.92) | 1.55 (0.89 to 2.69) |
| $OR_2$ (95% CI) | 1.0 (ref.) | 1.44 (0.56 to 3.72) | 1.53 (0.71 to 3.29) |

Logistic regression analyses; OR (95% CI).
$OR_1$ adjusted for age.
$OR_2$ adjusted for age, alcohol consumption, body mass index, cardiovascular and cerebrovascular disease, cholesterol, hypertension, physical activity and psychological stress.
The bold values are statistical significant, p<0.05

few collaterals.[38 39] When we used white matter lesions as a potential confounder between smoking and frontal lobe atrophy, the association between smoking and frontal lobe atrophy remained, that is, this potential confounder did not modify the association. One explanation could be that the study only measured periventricular lesions on CT and not the total subcortical lesions volume, as there was no such information available. The findings are, however, in line with several other studies that neither found associations between smoking and less white matter volume.[28 40 41]

Frontal lobe shrinkage is frequently found in cognitive healthy older people and can occur without clinically correlated symptoms.[20–23] In our sample, over half of the population had some degree of frontal lobe atrophy. Only a minority of these (8%) were diagnosed with dementia and none of them with frontotemporal dementia. However, frontal atrophy can give rise to cognitive and psychiatric symptoms, which affect a persons' functioning, behaviour and well-being.[42] Only a few studies have analysed the association between smoking and frontotemporal dementia, and none of them have found evidence of causality.[43–45]

In our study, only those who reported smoking in all three midlife examinations had higher attendance of frontal lobe atrophy, while no associations were found in temporary smokers. Our findings suggest that the effect of smoking on the frontal cortex decreases when smoking is stopped, as also has been shown for other adverse outcomes of smoking, such as cancer or cardiovascular disease.[46 47] Another observation was that the risks (ORs) were slightly higher in smokers in the 1980–1981 examination, compared with the 1968–1969 and 1974–1975 examinations, although not statistically significant. That can be a sign of an increased risk of smoking at higher ages, and that the women who smoked in 1980 mainly were the same as the longstanding smokers (ie, 81 of 88 persons).

The strengths of our study are the population-based sampling method, the long follow-up period with a prospective design, repeated and detailed information on smoking habits during early and late midlife and the

possibility to control for several potential confounders. However, some methodological issues need to be considered. First, we had no detailed data on smoking habits before or after the study period, and no data on the number of cigarettes/per day. Second, visual rating of brain atrophy and white matter lesions on CT is a rather crude method. CT scans are less sensitive than MRI to detecting brain structural changes, but similar in measuring atrophy.[48] Third, attrition is a problem in long-term prospective studies. Compared with non-participants, participants in the CT study were younger and had a lower mortality rate. They may thus be healthier and have less brain changes than the general population. If anything, this might likely have underestimated our findings. Fourth, we have no CT data from the midlife examinations (1968–1981) and can therefore not exclude that some of the women might have had cortical atrophy already in midlife and that early subtle changes in the frontal lobes may, for example, led to addiction habits, such as smoking. Fifth, there could be a potential stigma related to the self-reporting of cigarettes and alcohol use. If a lower prevalence were reported, this might have underestimated our findings. Sixth, our study focused on smoking. Other forms of nicotine use, for example, snuff or water pipe, were not measured. However, the prevalence of these latter habits should have been very low among middle-aged women between 1968 and 1981. Seventh, the sample size was sometimes too small for examining subgroups, for example, specific age cohorts. Finally, the study includes only Caucasian women. Thus, our results cannot be generalised to other populations.

## Conclusion

The study found an association between tobacco smoking in midlife and frontal lobe atrophy in late life. The strongest association was found in women with a higher alcohol consumption. Only longstanding smoking was associated with higher risk of frontal atrophy, not sporadic smoking. Destruction of synapses and neurons in the frontal cortex might be a consequence of long-term toxic stimulation of the nicotine receptors. The findings have both clinical and public health implications, and further studies should analyse if longstanding tobacco use also are related to impaired cognitive functions.

**Acknowledgements** The authors thank all members of the Prospective Population Study of Women group for their cooperation in data collection and management and Valter Sundh for statistical assistance.

**Contributors** All authors (LJ, XG, SS, MMF, SK, AZ and IS) contributed to the conception or design of the work and drafted and approved the final version. All agreed to be accountable for all aspects of the work in ensuring that questions related to the accuracy or integrity of any part of the work are appropriately investigated and resolved. LJ is guarantor for the study and responsible for the overall content.

**Funding** The study was financed by the ALF agreement (ALF 716681), Swedish Research Council for Health, Working Life and Wellfare (2004-0145, 2006-0596, 2008-1111, 2010-0870, 2012-1138, 2013-1202, 2018-00471, AGECAP 2013-2300), Hjärnfonden (FO2014-0207, FO2016-0214, FO2018-0214, FO2019-0163), The Alzheimer's Association Stephanie B. Overstreet Scholars (IIRG-00-2159), The Alzheimer's Association Zenith Award (ZEN-01-3151) and Swedish Research

Council (2005-8460, 2007-7462, 2012-5041, 2015-02830, 2019-01096, 2013-8717, NEAR 2017-00639).

**Competing interests** None declared.

**Patient and public involvement** Patients and/or the public were not involved in the design, or conduct, or reporting, or dissemination plans of this research.

**Patient consent for publication** Consent obtained from parent(s)/guardian(s).

**Ethics approval** This study involves human participants. This study received ethical approval by the Ethics Committee for Medical Research at the University of Gothenburg (Reference Ö402 99). All protocols were performed in accordance with the Declaration of Helsinki principles. Participants gave informed consent to participate in the study before taking part.

**Provenance and peer review** Not commissioned; externally peer reviewed.

**Data availability statement** Data are available upon reasonable request.

**ORCID iDs**
Lena Johansson http://orcid.org/0000-0002-0564-365X
Simona Sacuiu http://orcid.org/0000-0003-0472-7699

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
