## [Reviewer comments · BMJ Open]

This paper was submitted to a another journal from BMJ but declined for publication following peer review. The authors addressed the reviewers' comments and submitted the revised paper to BMJ Open. The paper was subsequently accepted for publication at BMJ Open.

ARTICLE DETAILS

TITLE (PROVISIONAL)	Longstanding smoking associated with frontal brain lobe atrophy: A 32-year follow-up study in women
AUTHORS	Johansson, Lena; Guo, Xinxin; Sacuiu, S; Fässberg, Madeleine; Kern, Silke; Zettergren, Anna; Skoog, Ingmar

VERSION 1 – REVIEW

REVIEWER	Ebmeier , Klaus University of Oxford, Psychiatry
REVIEW RETURNED	31-Mar-2023

GENERAL COMMENTS	Semiquantitative (typically 3-level scores) head x-ray CT study in 379 women examined lobar atrophy and periventricular white matter lesions at 76 years mean age after an average 32 years follow-up. Behavioural data, e.g., cigarette smoking, were available from baseline and two follow-up waves. Logistic regressions of smoking effects in midlife on brain scores were adjusted for alcohol consumption, vascular disorders, and several other potential confounders, There appeared to be a dose-dependent and a recency effect of smoking on frontal atrophy. A number of issues require clarification: 1. "An inter-observer agreement between the rating neurologist and a neuroradiologist was done in 130 CT-scans" - the ICCs given are quite low, and the derivation of ICCs and "concordances" are not further defined. The references given are not helpful, as they essentially contain identical text. Please define the formulae of the statistics used and the statistical programme, and reference these. Please interpret the magnitude of the ICCs and explain if and how you reconciled disagreements in scoring.2. Multiple analyses were done for different outcome measures Please describe how you controlled for false positive results. I liked the discussion which covered the crucial limitations of the paper. I can see that the primary hypothesis was of neurotoxic effects of smoking, but the reverse causality of frontal "atrophy" (which could have been present for many years previously, or in fact be a trait variable) causing dependent behaviour, could have explained both dose and recency effects, and the interaction with alcohol use!
--

REVIEWER	Samboju, Vishal
-----------------	-----------------

	University of California System
REVIEW RETURNED	04-May-2023

GENERAL COMMENTS	This work investigated the effect of smoking at different time points and the presence of brain atrophy (defined by physician) on CT scan. A cohort of 369 women over the course of years from 1968 to 1981 were analyzed. It was determined that there atrophy was associated in the group that was reported to be smoking at all three time intervals. I believe this work can benefit from revisions, I will write them in no specific order:  - if the authors were able to better define psychological stressors in terms of DSM criteria (for ex - they mention sleep disturbances - does this meet insomnia criteria? does nervousness correspond to someone diagnosed with generalized anxiety?) Otherwise the symptoms must be described as self-reported to more generalizeable. However, if the group has clinical data on underlying psychiatric disorders that are present they can report it as they do for blood pressure- i.e if there is a psychiatric disorder present or they are medicated. If this is too complex and not align with the research question per the author's intention, perhaps they could simply correlate psychological stress level. - although the authors do not have packs/ day for the patients, the amount of years smoking may be helpful. - what is the significance of deep versus shallow drag? why is it mentioned in the work? - regarding the measurement of smoking: the paper relays the measurement indexes for if people smoked in the year prior or are current smokers etc. However, it is reasonable to assume that if a 100 pack year smoker was a part of this study and not reported to be smoking at the times measured then they would not fit the study hypothesis. Thus it would be helpful to have some idea of how long if not how many cigarettes a person has smoked. - level of education may be helpful at it may relate to the degree of atrophy and is commonly known to be independent variable in brain reserve Cognitive function and frontal lobe atrophy in normal elderly adults: Implications for dementia not as aging-related disorders and the reserve hypothesis (meguro et al) - can the research team please further delineate the associations between alcohol consumption and frontal atrophy, is this a confound or is smoking truly the variable associated with atrophy. - physical activity can possibly be made more generalize able if intensity can be included. Not necessary, but if available. - did the people that developed dementia have ApoE status available or family history of dementia? - hypertensive urgency in the US commonly involves SBP > 160 and DBP > 100. If this is present, please state the distribution of BP ranges.
--

	- please further describe white matter lesions, perhaps cite further articles explaining clinical correlates and significance (seen in people with CVD, CAD, migraines, MS etc.) as it stands it simply described as lesion and unclear size, etiology etc. - can the research group make it more easily apparent to see at what average age the CT scans were completed? - did any of these participants have TBIs or other brain injury, other underlying psychiatric disease? In an ideal scenario, this paper could more clinically useful if MRI was performed and white matter lesions can be better identified. This work could hold more significance if smoking was better defined (in the U.S. pack years are commonly the unit used and also utilized in the USPTF guidelines for cancer screening). Along these same lines, it is not easy to generalize psychological stress as the work poorly defines this. However, it would be more beneficial if DSM or depression/ anxiety scales were used (if available)- as it stands this is a poorly quantified/described variable. Cholesterol would be useful in this instance or for more clinical value ASCVD risk (which is also widely used in clinical guidelines for practice in medicine). If the hypothesis aligns with the notion that smoking is related to the level of vascular dementia and frontal lobe atrophy- please further elucidate in discussion. More standardized dementia rating per the country of origin may also be used to describe the population sample and further provide confidence regarding the work's clinical integrity I am happy to re-review this work and look at the changes the team is willing/ amenable to make.
--	---

VERSION 1 – AUTHOR RESPONSE

Samboju, Vishal
University of California System

04-May-2023

This work investigated the effect of smoking at different time points and the presence of brain atrophy (defined by physician) on CT scan. A cohort of 369 women over the course of years from 1968 to 1981 were analyzed. It was determined that there atrophy was associated in the group that was reported to be smoking at all three time intervals.

I believe this work can benefit from revisions, I will write them in no specific order:

- if the authors were able to better define psychological stressors in terms of DSM criteria (for ex - they mention sleep disturbances - does this meet insomnia criteria? does nervousness correspond to someone diagnosed with generalized anxiety?) Otherwise the symptoms must be described as self-reported to more generalizeable. However, if the group has clinical data on underlying psychiatric disorders that are present they can report it as they do for blood pressure- i.e if there is a psychiatric disorder present or they are medicated. If this is too complex and not align with the research question per the author's intention, perhaps they could simply correlate psychological stress level.

- although the authors do not have packs/ day for the patients, the amount of years smoking may be helpful.

REFEREE 1

1. "An inter-observer agreement between the rating neurologist and a neuroradiologist was done in 130 CT-scans" - the ICCs given are quite low, and the derivation of ICCs and "concordances" are not further defined. The references given are not helpful, as they essentially contain identical text. Please define the formulae of the statistics used and the statistical programme, and reference these. Please interpret the magnitude of the ICCs and explain if and how you reconciled disagreements in scoring.

Response: Thank you for the comment. We agree with reviewer that the inter-rater information needed to be clarified. We have now included the Kappa values instead (page 6, paragraph 4). These analyses are used and described by the two CT-scan raters (of CT-scans) in earlier publication (Simoni M, Pantoni L, Pracucci G et al. Prevalence of CT-detected cerebral abnormalities in an elderly Swedish population sample. *Acta Neurol Scand* 2008;118(4),260-267)

2. Multiple analyses were done for different outcome measures. Please describe how you controlled for false positive results.

Response: We agree with the reviewer and have added new analyses. Correlations between the five outcome variables are tested and reported by Pearson correlation coefficient (95% CI), in additional file. Also, we multiplied the p value of the main finding (association between longstanding smoking and frontal lobe atrophy) with five (numbers of outcomes) to control for false positive result in multiple analyses ($p .003 \times 5 = p.015$) (page 10, paragraph 1).

I liked the discussion which covered the crucial limitations of the paper. I can see that the primary hypothesis was of neurotoxic effects of smoking, but the reverse causality of frontal "atrophy" (which could have been present for many years previously, or in fact be a trait variable) causing dependent behaviour, could have explained both dose and recency effects, and the interaction with alcohol use!

Response: We have now added text about reverse causality in the 'Strengths and limitations of this study' section (page 3) and in the Discussion (page 15, paragraph 2).

REFEREE 2

1. If the authors were able to better define psychological stressors in terms of DSM criteria (for ex - they mention sleep disturbances - does this meet insomnia criteria? does nervousness correspond to someone diagnosed with generalized anxiety?) Otherwise the symptoms must be described as self-reported to more generalizeable. However, if the group has clinical data on underlying psychiatric disorders that are present they can report it as they do for blood pressure- i.e if there is a psychiatric disorder present or they are medicated. If this is too complex and not align with the research question per the author's intention, perhaps they could simply correlate psychological stress level.

Response: Thank you for the careful reading and valuable reflections. However, we are sorry to say that we have no diagnoses for stress disorders in the dataset. Psychological stress was measured by a self-reported question that has been used in multiple publication. High levels of psychological stress have been found to be associated with e.g., vascular risk factors, neuroticism, life-stressors, and dementia. Given the broad concept of psychological stress, we choose to remain this variable in the study. Depression and anxiety are clinical disorders (the DSM manual were not accessible in the

1960s) and in population-based studies psychiatric disorders are reported in small numbers and gives low statistical power. We can include a discussion about these issues if editor decides to.

2. Although the authors do not have packs/day for the patients, the amount of years smoking may be helpful.

Response: We are sorry to say we does not have that this information.

3. What is the significance of deep versus shallow drag? why is it mentioned in the work?

Response: Our study use data from a multi-disciplinary research context, which includes a major number of tests and questions that can be used in different research projects. At time for the baseline examination, several researchers had focus on e.g., cardiovascular- and lung function disorders. For this purpose, detailed information about smoking habits were adequate, as for example, if smokers take deep drag. (Which cause a higher amount of nicotine in the blood stream.) We repeated the question-answers on smoking habits in this manus, even if the detailed information about deep vs. shallow drags are not used. In this study, we applied the answer alternatives 3-5 to define a 'current smoker' (3=Stopped smoking last 0-12 month; 4=Current smoking, but no deep drag; or 5=Current smoker, does deep drag). This categorization has been used in multiple former publications.

4. Regarding the measurement of smoking: the paper relays the measurement indexes for if people smoked in the year prior or are current smokers etc. However, it is reasonable to assume that if a 100 pack year smoker was a part of this study and not reported to be smoking at the times measured then they would not fit the study hypothesis. Thus it would be helpful to have some idea of how long if not how many cigarettes a person has smoked.

Response: We agree and are aware of this shortage, but unfortunately, we do not have this information. There is information in the limitation section about this issue (page 14, paragraph 2).

5. Level of education may be helpful at it may relate to the degree of atrophy and is commonly known to be independent variable in brain reserve. Cognitive function and frontal lobe atrophy in normal elderly adults: Implications for dementia not as aging-related disorders and the reserve hypothesis (meguro et al)

Response: We are sorry to say that we do not have information about the participants level of education.

6. Can the research team please further delineate the associations between alcohol consumption and frontal atrophy, is this a confound or is smoking truly the variable associated with atrophy.

Response: In our study, the alcohol consumption itself had no association with brain atrophy (page 10, paragraph 2). However, the levels of alcohol consumption were quite low, in this population of middle-aged women in the 1960s. For reaching statistical power, we grouped the participants in alcohol consumption, as drinking \leq once weekly vs. $>$ once weekly. We are aware of this very broad definition, and that we need to interpret the results with caution. So unfortunately, from this dataset, we cannot know if alcohol consumption is a mediator or moderator, or reverse causality. We write about this in the Limitations and Discussion (Page 3 and page 12, paragraph 2).

7. Physical activity can possibly be made more generalize able if intensity can be included. Not necessary, but if available.

Response: We are sorry to say that this is not measured at study examinations.

8. Did the people that developed dementia have ApoE status available or family history of dementia?

Response: The population study has shortcomings in information about ApoE status and family history of dementia (missing data). However, the number of dementia cases are low (n=17, <5%) and too small to be used in sub-analyses (e.g., smokers vs. non-smokers). Dementia is not the focus in this paper.

9. Hypertensive urgency in the US commonly involves SBP > 160 and DBP > 100. If this is present, please state the distribution of BP ranges.

Response: We have chosen to use the definition of 'Grade 1 hypertension' and also include those with antihypertensive medication (Hypertoni - Viss.nu)

10. Please further describe white matter lesions, perhaps cite further articles explaining clinical correlates and significance (seen in people with CVD, CAD, migraines, MS etc.) as it stands it simply described as lesion and unclear size, etiology etc.

Response: We have information about white matter lesions, i.e., regarding etiology, risk factors, comorbidity and cognitive symptoms, both in Introduction (page 4, paragraph 2) and Discussion (page 13, paragraph 2). Information on e.g., MS and migraines is not so adequate in this study aim.

11. Can the research group make it more easily apparent to see at what average age the CT scans were completed?

Response: We have now added detailed information about the participants ages in time for the CT-study, in Table 3.

12. Did any of these participants have TBIs or other brain injury, other underlying psychiatric disease?

Response: During the study period (32 years), several participants probably had brain trauma, or neurological and psychiatric disorders. The data about these conditions are unfortunately not complete. Also, the prevalence of these conditions is probable in small numbers, which give low statistical power in analyses.

In an ideal scenario, this paper could more clinically useful if MRI was performed and white matter lesions can be better identified. This work could hold more significance if smoking was better defined (in the U.S. pack years are commonly the unit used and also utilized in the USPTF guidelines for cancer screening). Along these same lines, it is not easy to generalize psychological stress as the work poorly defines this. However, it would be more beneficial if DSM or depression/anxiety scales were used (if available)- as it stands this is a poorly quantified/described variable. Cholesterol would be useful in this instance or for more clinical value ASCVD risk (which is also widely used in clinical guidelines for practice in medicine). If the hypothesis aligns with the notion that smoking is related to the level of vascular dementia and frontal lobe atrophy- please further elucidate in discussion. More standardized dementia rating per the country of origin may also be used to describe the population sample and further provide confidence regarding the work's clinical integrity

Response: Se answers above. If editor want us to describe this further, please let us know.

VERSION 2 – REVIEW

REVIEWER	Ebmeier , Klaus
----------	-----------------

	University of Oxford, Psychiatry
REVIEW RETURNED	07-Jul-2023

GENERAL COMMENTS	The authors have adequately addressed my comments.
--